

# Chemical composition, antioxidant and antitumor activities of sub-fractions of wild and cultivated *Pleurotus ferulae* ethanol extracts

Yi Yang[1], Changshuang Fu[1], Fangfang Zhou[1], Xiaoyu Luo[1], Jinyu Li[2], Jun Zhao[3], Jiang He[3], Xiaoqin Li[4] and Jinyao Li[1,4]

[1] College of Life Science and Technology, Xinjiang University, Urumqi, China
[2] College of Life Science, Xinjiang Normal University, Urumqi, China
[3] Key Laboratory for Uighur Medicine, Institute of Materia Medica of Xinjiang, Urumqi, China
[4] Affiliated Tumor Hospital of Xinjiang Medical University, Urumqi, China

Corresponding authors
Jinyu Li, lijinyu234@163.com
Jinyao Li, ljyxju@xju.edu.cn

## ABSTRACT

*Pleurotus ferulae* is an edible and medicinal mushroom with various bioactivities. Here, the ethanol extracts of wild and cultivated *P. ferulae* (PFEE-W and PFEE-C) and their subfractions including petroleum ether (Pe-W/Pe-C), ethyl acetate (Ea-W/Ea-C) and *n*-butanol (Ba-W/Ba-C) were prepared to evaluate their antioxidant and antitumor activities. Both PFEE-W and PFEE-C show the antioxidant activity and PFEE-W is stronger than PFEE-C. The antioxidant activities of their subfractions are in the following order: Ea > Ba > Pe. Moreover, PFEE-W and PFEE-C significantly inhibit the proliferation of murine melanoma B16 cells, human esophageal cancer Eca-109 cells, human gastric cancer BGC823 cells and human cervical cancer HeLa cells through induction of apoptosis, which partially mediated by reactive oxygen species. The antitumor activities of their subfractions are in the following order: Ea ≥ Pe > Ba. Pe-W shows higher antitumor activity compared with Pe-C, which might be correlated with the difference of their components identified by gas chromatography-mass spectrometry. These results suggest that both wild and cultivated *P. ferulae* have antioxidant and antitumor activities, and cultivated *P. ferulae* could be used to replace wild one in some functions.

# INTRODUCTION

Currently, cancer is the second leading cause of death in the world. In 2015, there were 17.5 million cancer cases and 8.7 million deaths globally, which included around 4.3 million cancer cases and 2.8 million deaths in China (*Chen et al., 2016*; *Global Burden of Disease Cancer Collaboration et al., 2017*). Different strategies including surgery, chemotherapy, radiotherapy, hormone therapy, targeted therapy or their combinations have been developed to treat cancer (*Chang et al., 2015*; *Eberhardt et al., 2015*; *Le Tourneau et al., 2015*; *Perez et al., 2014*).

Medicinal fungi have various biological functions and become a hot spot in functional food and medical research. *Pleurotus ferulae* is an edible and medicinal

mushroom which grows on the living rhizome trunks of *Ferula asafoetida* in the Gobi desert and mainly distributed in Xinjiang, China (*Wang et al., 2014*). Various biological components including fibrinolytic enzyme, lectin, pleurone and polysaccharides have been isolated from *P. ferulae* (*Choi et al., 2017*; *Xu et al., 2014*; *Lee et al., 2011*; *Li et al., 2017*). A growing body of research including ours has reported that *P. ferulae* extracts and some components show antioxidant, antihyperlipidemic, antitumor, antimicrobial and immunoregulatory effects (*Alam et al., 2012*; *Choi et al., 2004*; *Kalyoncu et al., 2010*; *Li et al., 2015a*; *Wang et al., 2014*). *Alam et al. (2012)* have shown that the acetonic and methanolic extracts of *P. ferulae* exhibit better antioxidant activities than hot water extracts. *Choi et al. (2004)* have reported that *P. ferulae* ethanol extracts show stronger cytotoxicity against A549 cells than hot water extracts. Recently, we compared the antitumor activities of wild and cultivated *P. ferulae* ethanol extracts (PFEE-W and PFEE-C). Although PFEE-W exhibited higher antitumor activity than PFEE-C, both PFEE-C and PFEE-W significantly inhibited the growth of hepatocellular carcinoma cells through induction of apoptosis (*Yang et al., 2018*). Due to the fact that the source of wild *P. ferulae* is scarce, it has been successfully domesticated by Xinjiang Institute of soil biological desert in 1990. However, further investigation is needed to determine whether wild and cultivated *P. ferulae* have similar or different antioxidant activities and antitumor effects on different types of tumors.

In the present study, PFEE-W and PFEE-C were prepared and their major components were analyzed. PFEE-W and PFEE-C were further extracted by petroleum ether, ethyl acetate and *n*-butanol to collect the corresponding subfractions and named as Pe-W/Pe-C, Ea-W/Ea-C and Ba-W/Ba-C, from which antioxidant and antitumor activities were evaluated. Then the fatty acid compositions of Pe-W/Pe-C were identified by gas chromatography-mass spectrometry (GC-MS). We found that PFEE-W and PFEE-C and their subfractions possessed antioxidant activity and could inhibit the proliferation of tumor cells through induction of cell apoptosis and necrosis.

## MATERIAL AND METHODS

### Preparation of wild and cultivated *P. ferulae* extractions

Cultivated and wild *P. ferulae* were collected from Jinghe in Xinjiang Uygur Autonomous Region, China. Wild and cultivated *P. ferulae* ethanol extracts (PFEE-W and PFEE-C) were prepared according to our previous protocol (*Yang et al., 2018*). The stepwise extractions of PFEE-W and PFEE-C were performed using petroleum ether, ethyl acetate and *n*-butanol orderly to obtain six subfractions and named as Pe-W/Pe-C, Ea-W/Ea-C and Ba-W/Ba-C, respectively. These subfractions were concentrated using a rotary vacuum evaporator at 45 °C. After air drying to remove the solvents, the six subfractions were dissolved in 100% dimethyl sulfoxide (DMSO) (Sigma-Aldrich, St. Louis, MO, USA) at the concentration of 100 mg/ml and sterilized with a 0.22 μm filter.

### Determination of polysaccharide content

The polysaccharide contents of PFEE-W/C or subfractions were determined using the phenol–sulfuric acid method according to previous protocol (*DuBois et al., 1956*).

The optical density (OD) was measured at 490 nm using a 96-well microplate reader (Bio-Rad Laboratories, Hercules, CA, USA). The polysaccharide content was calculated according to the standard curve made by the standard of glucose.

## Determination of polyphenol content

The polyphenol contents of PFEE-W/C or subfractions were determined by ferrous tartrate method (*Yu et al., 2007*). The OD was measured at 540 nm using a 96-well microplate reader. The polyphenol content was calculated according to the standard curve made by the standard of gallic acid.

## Determination of total flavonoid content

The flavonoid contents of PFEE-W/C or subfractions were detected according to previous description (*Swamy, Sinniah & Akhtar, 2015*). The OD was detected at 517 nm using a 96-well microplate reader. The content of flavonoids was calculated according to the standard curve obtained by the standard of rutin.

## Diphenylpicrylhydrazyl radical scavenging assay

Diphenylpicrylhydrazyl (DPPH) radical scavenging assay was carried out according to previously described (*Molyneux, 2004*). Briefly, PFEE-W/C or subfractions were dissolved in ethanol and mixed with 0.1 mM DPPH. The ascorbic acid was used as positive control and the mixture of ethanol and DPPH was used as negative control. The OD was measured at 517 nm using a microplate reader. Tests were carried out in triplicate. The percentage of DPPH radical scavenging was calculated according to the equation:

$$\text{DPPH radical scavenging } (\%) = \left(1 - \frac{A_s - A_c}{A_b}\right) \times 100\%$$

$A_s$ is the OD of the sample mixed with DPPH, $A_c$ is the OD of sample without DPPH and $A_b$ is the OD of the ethanol mixed with DPPH.

## Reducing power

The reducing activities of PFEE-W/C or subfractions were carried out according to previous description (*Yildirim, Mavi & Kara, 2001*). The ascorbic acid was used as positive control. The OD was measured at 700 nm.

## Cell lines and cell culture

Murine melanoma B16 cells, human esophageal cancer Eca-109 cells, human gastric cancer BGC823 cells, human cervical cancer HeLa cells and mouse liver NCTC1469 cells were obtained from the Xinjiang Key Laboratory of Biological Resources and Genetic Engineering in Xinjiang University (Urumqi, Xinjiang, China) and cultured in RPMI 1640 medium (Gibco, Waltham, MA, USA) supplemented with 10% heat-inactivated fetal bovine serum (MRC, Jiangsu, China), 100 U/ml penicillin and 100 μg/ml streptomycin at 37 °C in a humidified environment with 5% $CO_2$.

## MTT assay

MTT [3-(4,5-dimethyl-2-thiazolyl)-2,5-diphenyl-2-H-tetrazolium bromide]
(Sigma-Aldrich, St. Louis, MO, USA) assay was used to evaluate the effects of PFEE-W/C
and subfractions on the proliferation of tumor cells. The tumor cells at the logarithmic
phase of growth were seeded in 96-well plates at a density of $5 \times 10^3$ cells/well and
cultured overnight, then treated with PFEE-W/C or subfractions at various concentrations
(0, 100, 200, 400, 600 μg/ml) or 0.6% DMSO (equal to that in 600 μg/ml) for 24 h.
Cisplatin was used as a positive control. After centrifugation at 1,200 rpm for 7 min,
supernatant was discarded and 100 μl of MTT solution (0.5 mg/ml in PBS) was added into
each well and incubated at 37 °C for 4 h. The formed formazan crystals were dissolved
in 150 μl DMSO. The OD490 values were measured by a 96-well microplate reader.
The relative cell viability was calculated as the followed formula: Cell viability (%) =
$(OD_{treated}/OD_{untreated}) \times 100\%$.

## Analysis of cell apoptosis

HeLa cells were treated with PFEE-W/C and subfractions at concentration of
400 μg/ml or 0.4% DMSO for 24 h. Cells were collected and stained with Annexin
V-FITC/propidiumidide (PI) Apoptosis Detection Kit (YEASEN, Shanghai, China),
China) according to the manufacturer's instructions. In some experiments, cells were
pretreated with 10 mM $N$-acetyl-L-cysteine (NAC, Sigma-Aldrich, St. Louis, MO, USA) for
2 h, and treated with 400 μg/ml Ea-W or Ea-C for 24 h to detect the apoptosis. Samples
were analyzed by flow cytometry (FACSCalibur, BD Biosciences, San Jose, CA, USA).

## Hoechst 33342 staining

After treatment with 400 μg/ml PFEE-W/C and subfractions or 0.4% DMSO for 24 h,
HeLa cells were washed with PBS and fixed with 4% ice–cold paraformaldehyde at 4 °C for
10 min. Then cells were stained with Hoechst 33342 (Beyotime, Shanghai, China) at 4 °C
for 10 min in the dark after washing with PBS. Samples were observed by inverted
fluorescence microscope (Nikon Eclipse Ti-E, Tokyo, Japan).

## Measurement of reactive oxygen species

Intracellular reactive oxygen species (ROS) production was measured using DCFH-DA
probes. HeLa cells were treated with 400 μg/ml Ea-W/Ea-C or 0.4% DMSO for 24 h.
After treatment, all cells were harvested and stained by 10 mM of fluorescent probe
DCFH-DA (Beyotime, Shanghai, China) for 20 min at 37 °C. After washing three times
with PBS, the fluorescence intensity in cells was determined by flow cytometry.

## Gas chromatography-mass spectrometry

The samples were prepared according to the following procedure. Briefly, 50 mg
Pe-W/Pe-C were mixed with one ml methanol and 50 μl concentrated sulfuric acid,
followed by reflux for 4 h. After cooling, saturated NaCl solution was add to terminate the
reaction. Then, the solution was extracted with two ml $n$-hexane, and the upper layer
was collected for GC-MS analysis. The methyl derivatives of fatty acids were identified

by a 7890A-5975C gas chromatograph equipped with a PE-5MS capillary column (30 m × 0.25 mm × 0.25 μm). Helium (flow rate, 1.0 ml/min) was used as the carrier gas. The column temperature was maintained initially at 100 °C for 1 min, followed by increasing to 200 °C at a rate of 3 °C/min, from 200 to 250 °C at a rate of 5 °C/min, and then kept at 250 °C for 30 min. The electron impact (EI) energy was 70 eV and the ion source temperature was 230 °C. Split ratio was 50:1. EI mass spectra were recorded in the 33–450 amu range at 1 s intervals. The library search was carried out using NIST GC-MS libraries.

## Statistical analysis

We expressed all data as mean ± standard error of the mean. Statistical analysis was performed by GraphPad Prism 5.0, and conducted with one-way analysis of variance. The two-tailed paired $t$-test was used to compare PFEE-W and PFEE-C and their subfractions. $p < 0.05$ was considered to be statistically significant.

## RESULTS

### The polysaccharide, polyphenol and flavonoid contents of PFEE-W and PFEE-C and subfractions

PFEE-W/C and subfractions were prepared to analyze the contents of polysaccharides, polyphenols and flavonoids. Compared with PFEE-C, PFEE-W contained higher concentration of flavonoids (1.202 vs 1.04 mg/ml), lower concentration of polysaccharides (38.46 vs 54.87 mg/ml) and similar concentration of polyphenols (Table 1). The polysaccharide and polyphenol contents in Pe-W are lower than that in Pe-C, while the flavonoid contents in Pe-W and Pe-C are similar. Ea-W and Ea-C contain similar polysaccharide contents but Ea-W contains higher concents of polyphenols and flavonoids than that of Ea-C. Same as Ea-W/Ea-C, Ba-W and Ba-C also contain similar polysaccharide contents, while Ba-W contains higher contents of polyphenols and flavonoids than that of Ba-C.

### Antioxidant activities of PFEE-W/C and subfractions

The antioxidant activities of PFEE-W/C and subfractions were measured by DPPH radical scavenging assay. As shown in Figs. 1A–1D, all fractions showed remarkable radical scavenging activities in a dose-dependent manner. Generally, the radical scavenging activities of fractions from wild $P.$ $ferulae$ are higher than that of cultivated $P.$ $ferulae$. The scavenging activity of ethyl acetate fraction was higher than those of petroleum ether and $n$-butanol fractions both in wild and cultivated $P.$ $ferulae$. However, the scavenging activities of PFEE-W/C and subfractions were lower than that of $V_c$, a positive control, which reached the 100% scavenging rate at one mg/ml.

The antioxidant activities of all fractions were further evaluated by reducing power. Similarly, all fractions showed reducing power in a dose-dependent manner, and $V_c$ showed higher reducing power than that of PFEE-W/C and subfractions (Figs. 1E–1H). The reducing power of fractions from wild $P.$ $ferulae$ is slightly higher than that of cultivated $P.$ $ferulae$. In addition, the reducing power of fractions from both wild and

**Table 1 The contents of polysaccharides, polyphenols and flavonoids in PFEE-W and PFEE-C and their subfractions.**

| Sample (100 mg/ml) | Polysaccharides (mg/ml) | Polyphenols (mg/ml) | Flavonoids (mg/ml) |
|---|---|---|---|
| PFEE-W | 38.46 ± 1.005[b] | 0.256 ± 0.011[a] | 1.202 ± 0.022[a] |
| Pe-W | 9.764 ± 0.618[e] | 0.099 ± 0.005[c] | 0.206 ± 0.002[d] |
| Ea-W | 20.36 ± 0.520[cd] | 0.150 ± 0.004[b] | 1.269 ± 0.037[a] |
| Ba-W | 40.64 ± 0.646[b] | 0.063 ± 0.001[c] | 0.609 ± 0.019[c] |
| PFEE-C | 54.87 ± 0.840[a] | 0.250 ± 0.017[a] | 1.04 ± 0.018[b] |
| Pe-C | 20.84 ± 1.011[c] | 0.211 ± 0.010[a] | 0.176 ± 0.006[de] |
| Ea-C | 16.14 ± 0.906[d] | 0.063 ± 0.006[c] | 0.566 ± 0.015[c] |
| Ba-C | 38.51 ± 1.143[b] | 0.009 ± 0.001[d] | 0.100 ± 0.002[e] |

Note:

Values in the column followed by a different letter superscript were significantly different ($p < 0.05$) and values had the same letters are not statistically significant ($p > 0.05$).

cultivated *P. ferulae* followed the order: PFEE > Ea ≥ Ba > Pe. These results indicate that PFEE-W/C and subfractions have antioxidant activities.

## PFEE-W/C and subfractions suppress the growth of tumor cells

The antitumor effect of PFEE-W/C and subfractions was detected and compared by MTT assay. B16, Eca-109, BGC823 and HeLa cells were treated with different concentrations of PFEE-W/C and subfractions for 24 h. As shown in Fig. 2, PFEE-W/C and subfractions significantly reduced the viability of B16, Eca-109 and HeLa cells in a dose-dependent manner except BGC823 cells that were not sensitive. PFEE-W and PFEE-C showed similar antitumor activities against B16 and Eca-109 cells but the antitumor activity of PFEE-W was higher than that of PFEE-C for HeLa cells. Furthermore, subfractions from wild and cultivated *P. ferulae* showed different antitumor activities. For Eca-109 and HeLa cells, Pe-W and Ea-W exhibited higher antitumor activities than that of Pe-C and Ea-C, respectively. The antitumor activities of subfractions from both wild and cultivated *P. ferulae* followed the order: Ea ≥ Pe > Ba. These results indicate that PFEE-W/C and subfractions have antitumor activities.

We also detected the cytotoxicity of PFEE-W/C and subfractions on mouse liver NCTC1469 cells. Generally, the low concentrations of PFEE-W/C and subfractions showed no cytotoxicity, while the high concentrations of PFEE-W/C and subfractions significantly reduced the viability of NCTC1469 cells (Fig. S1). However, the cytotoxicity of PFEE-W/C and subfractions on NCTC1469 cells was much lower than that on tumor cells, suggesting that PFEE-W/C and subfractions have minor side effect on normal cells.

## Chemical composition of fatty acids of Pe-W and Pe-C

In order to find out the components caused the different antitumor activities, Pe-W and Pe-C were selected to do GC-MS due to their much difference of antitumor activities. The fatty acid compositions of the two subfractions were identified by GC-MS (Fig. S2). A total of 21 components were identified in Pe-W (Table 2), which mainly contained methyl linoleate (60.95%), hexadecanoic acid methyl ester (10.44%), 11-octadecenoic acid

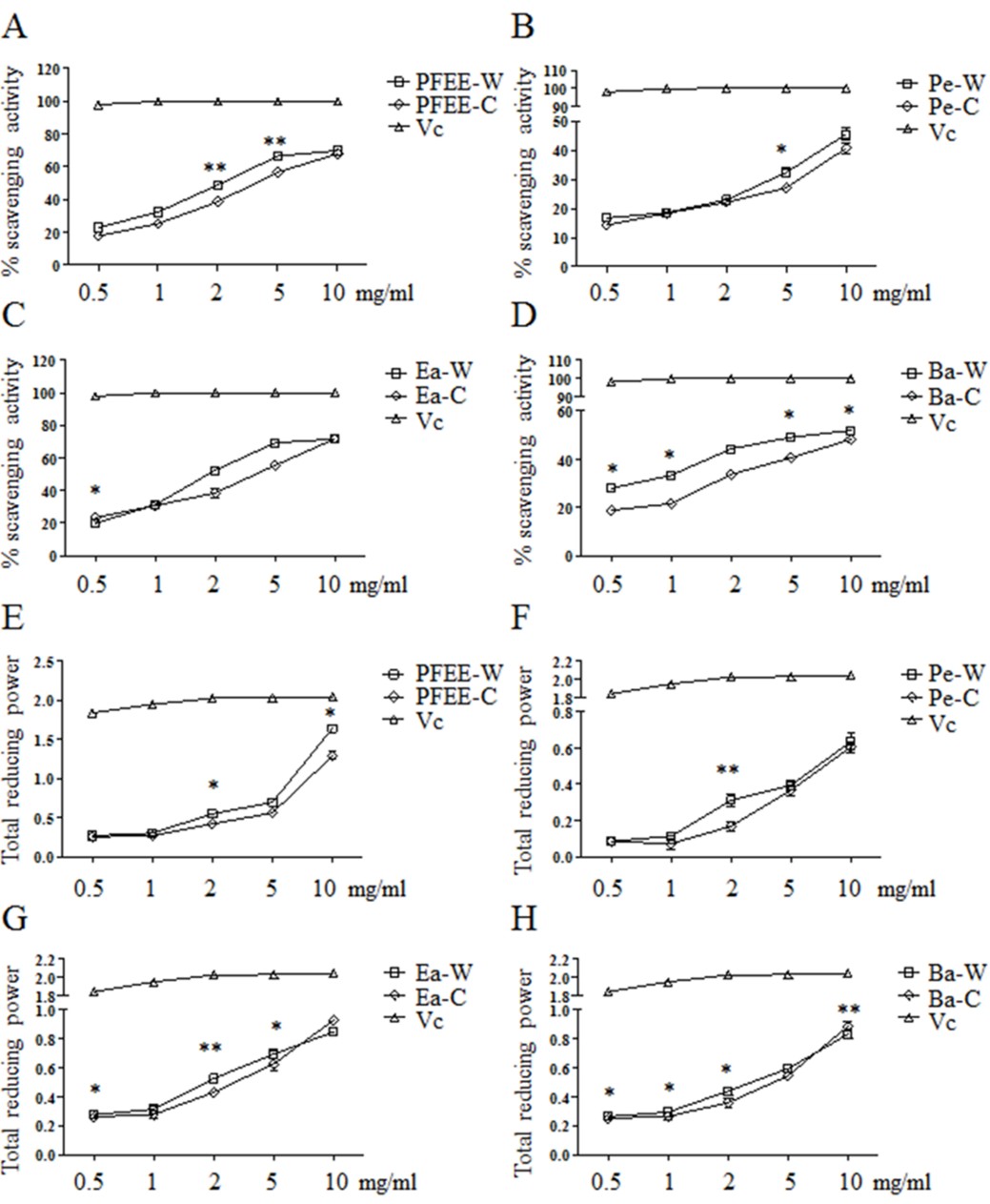

**Figure 1 The antioxidant activities of PFEE-W/C and their subfractions.** (A–D) DPPH scavenging activities. (E–H) Total reducing power. *Vc* was used as positive control. Data are from three independent experiments, the two-tailed paired *t*-test was used to compare wild and cultivated *P. ferulae* extracts. *$p < 0.05$; **$p < 0.01$.

methyl ester (6.95%) and hexadecanoic acid (4.1%). A total of 25 components were identified in Pe-C (Table 2), which mainly contained methyl linoleate (58.25%), methyl oleate (15.23%), hexadecanoic acid methyl ester (14.32%) and myo-inositol hexaacetate (6.15%). Compared the major constituents in Pe-W and Pe-C, the contents of methyl linoleate and hexadecanoic acid methyl ester are similar but the contents of 11-octadecenoic acid methyl ester and methyl stearate are different. Except for the quantitative differences of some components, the qualitative differences were also found

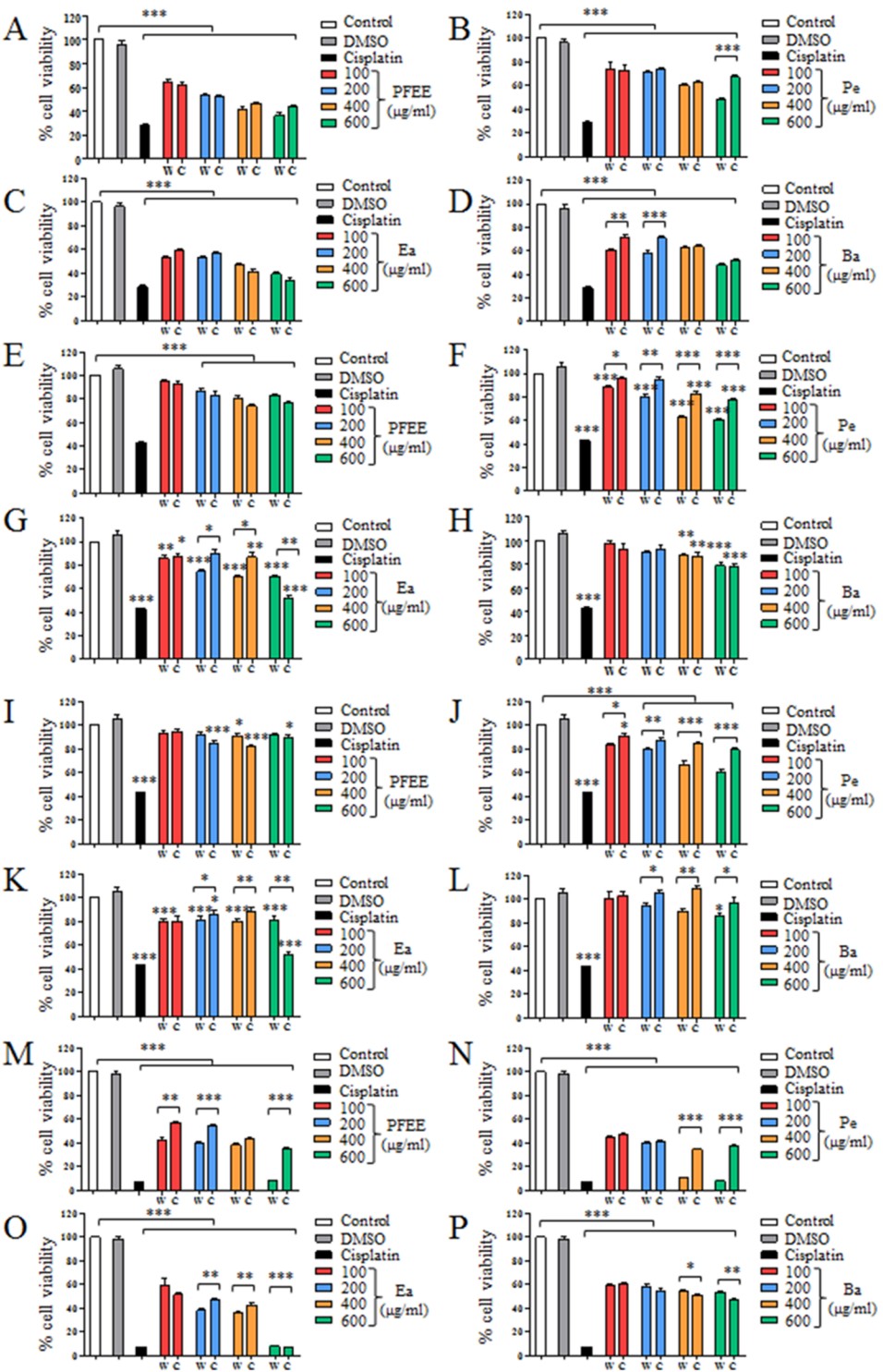

**Figure 2 Effect of PFEE-W/C and their subfractions on the growth of tumor cells.** The viability of B16 (A–D), Eca-109 (E–H), BGC823 (I–L) and HeLa (M–P) cells after treatment with PFEE-W/C and their sub-fractions for 24 h. Data are from three independent experiments and analyzed by ANOVA. $*p < 0.05$; $**p < 0.01$; $***p < 0.001$ compared to untreated group. The two-tailed paired $t$-test was used to compare wild and cultivated *P. ferulae* extracts. $*p < 0.05$; $**p < 0.01$; $***p < 0.001$.

**Table 2  Chemical composition and contents of fatty acids of Pe-W and Pe-C.**

| No. | Compounds | Retention time (min) | | Composition (%) | |
|---|---|---|---|---|---|
| | | Pe-W | Pe-C | Pe-W | Pe-C |
| 1 | Methyl tetradecanoate | – | 21.46 | – | 0.39 |
| 2 | Pentadecanoic acid, methyl ester | 26.60 | 26.09 | 0.76 | 0.73 |
| 3 | 9-Hexadecenoic acid, methyl ester | – | 29.49 | – | 0.65 |
| 4 | 7-Hexadecenoic acid, methyl ester | 30.03 | 29.96 | 0.24 | 0.16 |
| 5 | Hexadecanoic acid, methyl ester | 31.29 | 30.87 | 10.44 | 14.32 |
| 6 | Hexadecanoic acid | 33.03 | – | 4.10 | – |
| 7 | Octadecanoic acid | 33.39 | – | 0.13 | – |
| 8 | Hexadecanoic acid, 2-hydroxy-, methyl ester | 36.10 | – | 1.24 | – |
| 9 | Myo-inositol, hexaacetate | – | 36.72 | – | 6.15 |
| 10 | Methyl linoleate | 37.98 | 37.70 | 60.95 | 58.25 |
| 11 | Methyl oleate | – | 37.97 | – | 15.23 |
| 12 | 11-Octadecenoic acid, methyl ester | 38.25 | 38.06 | 6.95 | 0.73 |
| 13 | 13-Octadecenoic acid, methyl ester | 38.44 | – | 0.83 | – |
| 14 | Methyl stearate | 39.35 | 38.91 | 2.89 | 0.93 |
| 15 | 12-Methyl-E,E-2,13-octadecadien-1-ol | 39.62 | – | 2.27 | – |
| 16 | Z,E-3,13-Octadecadien-1-ol | 40.61 | – | 1.20 | – |
| 17 | Methyl 6-*cis*,9-*cis*,11-*trans*-octadecatrienoate | – | 42.51 | – | 0.24 |
| 18 | 2-Methyl-E,E-3,13-octadecadien-1-ol | – | 43.03 | – | 0.22 |
| 19 | Methyl 8,11,14,17-eicosatetraenoate | 43.48 | 43.46 | 0.64 | 0.20 |
| 20 | *cis*-5,8,12-Eicosatrienoic acid, methyl ester | – | 43.58 | – | 0.22 |
| 21 | 8,11-Eicosadienoic acid, methyl ester | – | 44.13 | – | 0.13 |
| 22 | *cis*-11-Eicosenoic acid, methyl ester | – | 44.39 | – | 0.14 |
| 23 | 13-Docosenoic acid, methyl ester, (Z)- | – | 48.75 | – | 0.10 |
| 24 | Bis(2-ethylhexyl) phthalate | – | 49.00 | – | 0.11 |
| 25 | Docosanoic acid, methyl ester | – | 49.19 | – | 0.08 |
| 26 | Cyclopropaneoctanoic acid, 2-octyl-, methyl ester | 49.42 | 49.38 | 0.13 | 0.07 |
| 27 | 11-Eicosenoic acid, methyl ester | – | 50.15 | – | 0.08 |
| 28 | 15-Tetracosenoic acid, methyl ester, (Z)- | 52.02 | 51.69 | 0.35 | 0.46 |
| 29 | Tetracosanoic acid, methyl ester | 52.51 | 52.17 | 0.22 | 0.18 |
| 30 | Butanedioic acid, 2,3-bis(8-nonen-1-yl)-, dimethyl ester | – | 53.47 | – | 0.09 |
| 31 | 12-Methyl-E,E-2,13-octadecadien-1-ol | 54.42 | – | 0.19 | – |
| 32 | Methyl 2-hydroxy-tetracosanoate | 55.04 | – | 0.61 | – |
| 33 | Methyl 17-hexacosenoate | – | 55.63 | – | 0.14 |
| 34 | (22Z)-Cholesta-5,7,22-trien-3-ol | 61.62 | – | 1.50 | – |
| 35 | Desmosterol | 63.32 | – | 2.63 | – |
| 36 | Ergosterol | 64.00 | – | 1.73 | – |

between Pe-W and Pe-C. Some components including hexadecanoic acid, 12-methyl-E, E-2,13-octadecadien-1-ol, desmosterol, ergosterol were found in Pe-W but not in Pe-C. Other components including methyl oleate and myo-inositol hexaacetate were found

in Pe-C but not in Pe-W. The difference of components might cause the different antitumor activities of Pe-W and Pe-C.

## PFEE-W/C and subfractions induce apoptosis of HeLa cells

To investigate whether PFEE-W/C and subfractions can induce apoptosis of tumor cells, HeLa cells were selected and treated with 400 μg/ml of PFEE-W/C, Pe-W/C and Ea-W/C for 24 h. After Annexin V-FITC and PI staining, cell apoptosis was analyzed by flow cytometry. Compared with untreated group, the frequencies of apoptotic and necrotic HeLa cells were significantly increased after PFEE-W/C, Pe-W/C and Ea-W/C treatment, but these extracts mainly induced apoptosis (>60%) compared with necrosis (<20%) (Figs. 3A–3C). PFEE-W and PFEE-C have similar activities in the induction of apoptosis and necrosis. Pe-W and Ea-W show higher activities than Pe-C and Ea-C, respectively, in the induction of apoptosis and necrosis. Furthermore, Hoechst 33342 staining was used to observe the morphology of nuclei of HeLa cells upon PFEE-W/C and subfractions treatment. As shown in Figs. 3D and 3E, the nuclei of untreated cells were homogeneously stained, but the nuclei of PFEE-W/C and subfractions treated cells showed condensed and fragmented. These results indicated that PFEE-W/C and subfractions induced apoptosis of HeLa cells.

## ROS partially mediates the induction of apoptosis by Ea-W and Ea-C

Reactive oxygen species plays an important role in the induction of apoptosis (*Redza-Dutordoir & Averill-Bates, 2016*). Due to the high antitumor activity, Ea-W and Ea-C were selected to investigate the role of ROS in the induction of apoptosis. After treatment for 24 h, Hela cells were stained with the specific fluorescence probe DCFH-DA to detect intracellular ROS levels. As shown in Figs. 4A and 4B, ROS production showed dynamically changes upon Ea-W and Ea-C treatment. ROS production was significantly increased at 2 h and gradually decreased from 8 to 24 h, then dramatically increased at 48 h.

Next, NAC, a ROS scavenger, was used to test the role of ROS in the induction of apoptosis. As shown in Figs. 4C–4E, NAC pretreatment partially inhibited the apoptosis of HeLa cells induced by Ea-W and Ea-C, suggesting that ROS produced during the first 2 h promoted the induction of apoptosis.

## DISCUSSION

Several studies including ours have been reported that *P. ferulae* contains a number of active components such as polysaccharides, flavonoids and polyphenols, and exhibits antioxidant, antitumor, antihyperglycemic and immunoregulatory activities (*Choi et al., 2004*, *2016*; *Alam et al., 2012*, *Li et al., 2015a*, *2017*; *Wang et al., 2014*). However, cultivated *P. ferulae* has been used in almost studies including ours due to the resource of wild *P. ferulae* is scarce. It is still elusive whether wild and cultivated *P. ferulae* have similar or different activities of antioxidation and antitumor. Here, the ethanol extracts of wild and cultivated *P. ferulae* and their subfractions were prepared to compare their components and antioxidant and antitumor activities. The results show that PFEE-C

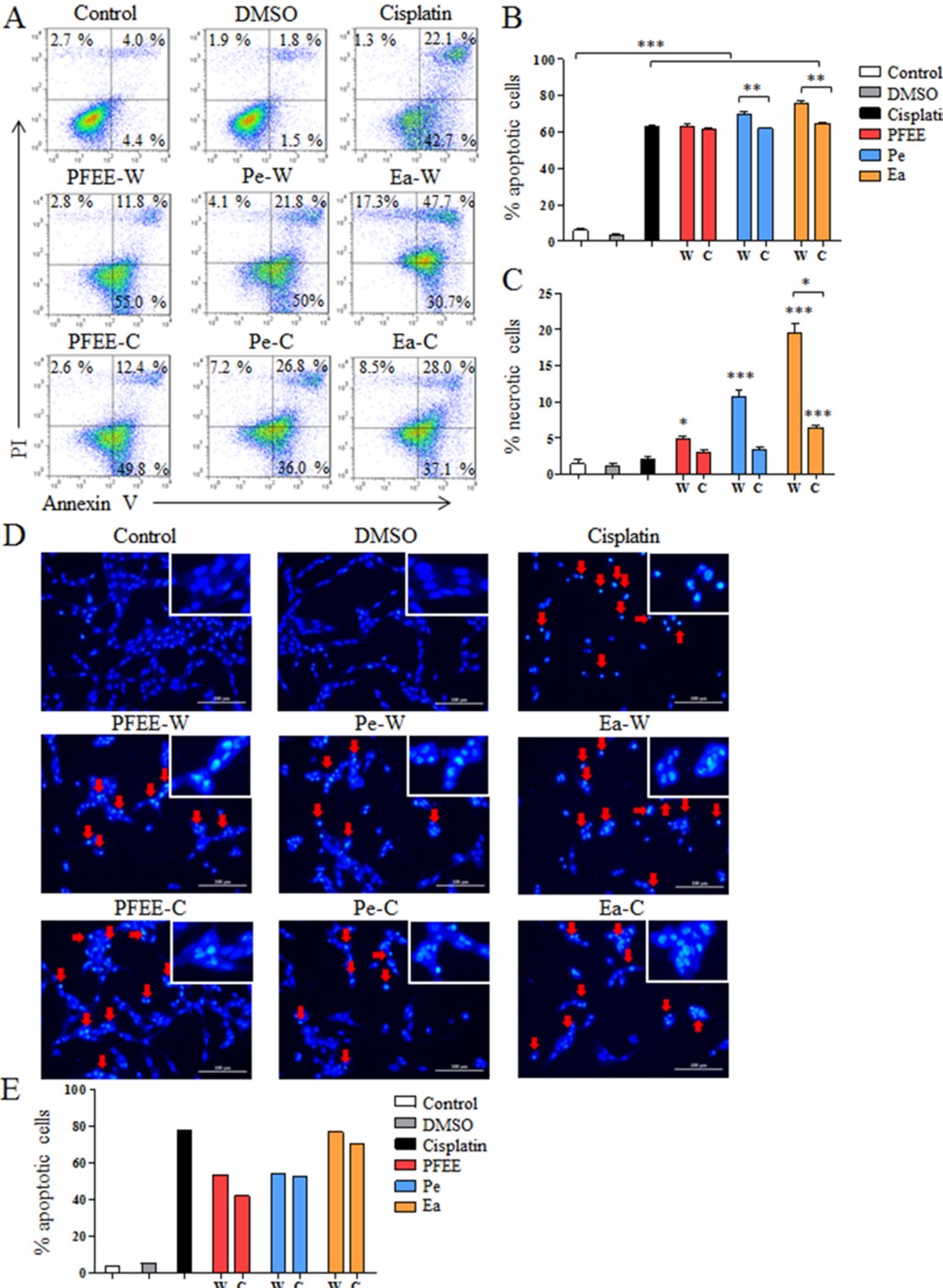

**Figure 3 The apoptosis of HeLa cells induced by PFEE-W/C and their subfractions.** A total of 400 µg/ml of PFEE-W/C and their subfractions were used to treat HeLa cells for 24 h. HeLa cells were stained by Annexin V/PI and analyzed by flow cytometry. (A) showed the individual dot plots. (B) and (C) showed the summary data. Data are from three independent experiments and analyzed by ANOVA. *$p < 0.05$; **$p < 0.01$; ***$p < 0.001$ compared to untreated group. The two-tailed paired $t$-test was used to compare wild and cultivated *P. ferulae* extracts, *$p < 0.05$; **$p < 0.01$. (D) The nuclear morphology of HeLa cells. HeLa cells were stained with Hoechst 33342 and observed by inverted fluorescence microscopy. The arrows indicated the chromosomal condensation. (E) showed the summary data.

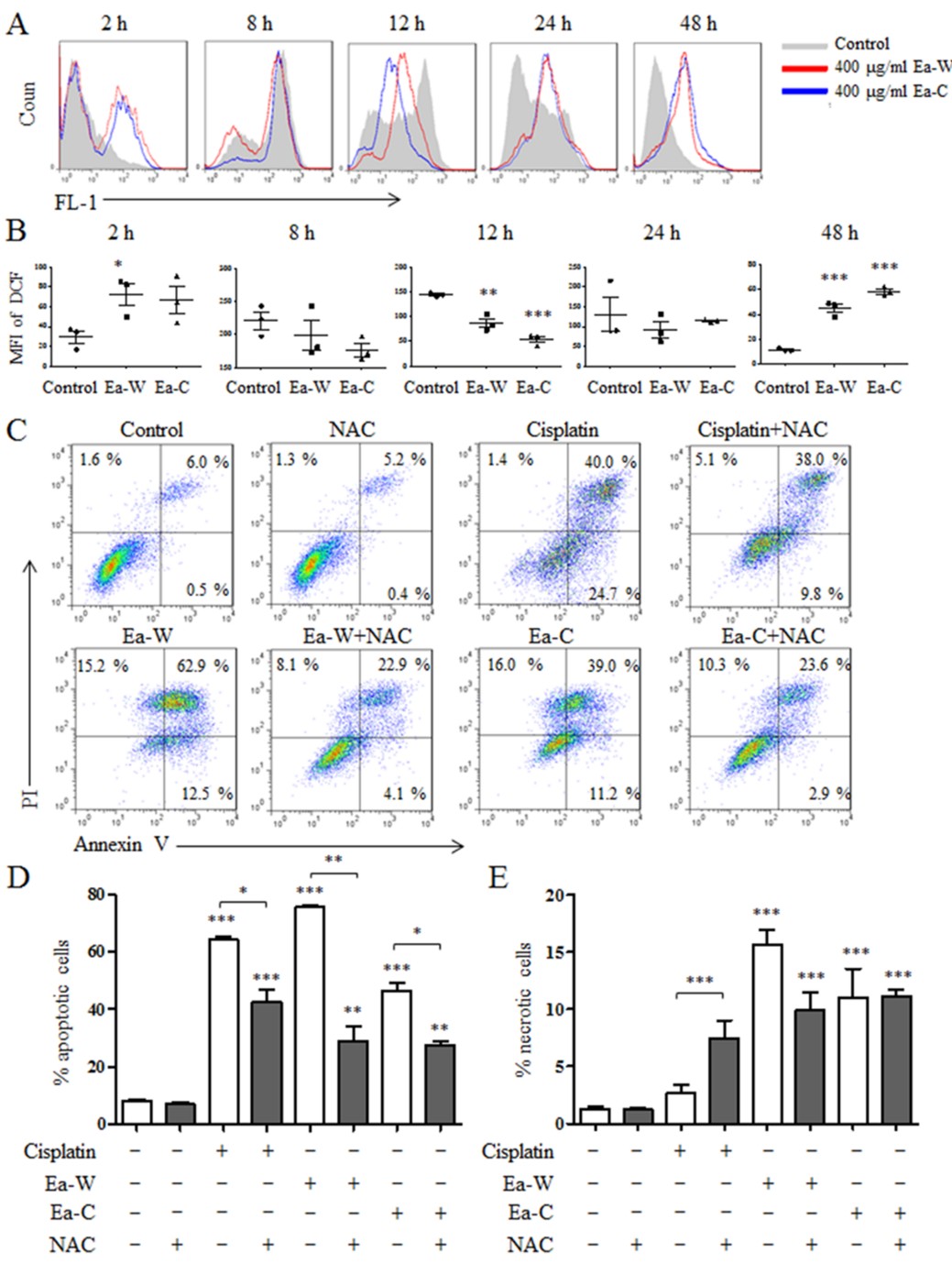

**Figure 4 Intracellular ROS production and its role in the induction of apoptosis upon Ea-W and Ea-C treatment.** (A) Intracellular ROS production in HeLa cells treated with 400 μg/ml Ea-W and Ea-C. After treatment for 2, 8, 12, 24 and 48 h, cells were stained with fluorescent probe DCFH-DA and analyzed by flow cytometry. (B) showed the summary data. (C) The role of ROS in the induction of apoptosis. HeLa cells were pretreated with 10 mM NAC for 2 h and treated with Ea-W and Ea-C for 24 h. After staining with Annexin V/PI, samples were analyzed by flow cytometry. (D) and (E) showed the summary data. Data are from three independent experiments and analyzed by ANOVA. $^*p < 0.05$; $^{**}p < 0.01$; $^{***}p < 0.001$ compared to untreated group. The two-tailed paired $t$-test was used to compare wild and cultivated $P.$ $ferulae$ extracts. $^*p < 0.05$; $^{**}p < 0.01$; $^{***}p < 0.001$.

contains higher concentration of polysaccharides and lower concentration of flavonoids than PFEE-W. The contents of polyphenols are similar in PFEE-W and PFEE-C. The contents of these components are different in their subfractions. Generally, PFEE-W/C and their subfractions show strong antioxidant and antitumor activities. Compared with PFEE-C and its subfractions, PFEE-W and its some subfractions showed higher antioxidant and antitumor activities to a certain extent, which might be correlated with the differences of their components.

Due to the much difference of antitumor activities between Pe-W and Pe-C, the components of Pe-W and Pe-C were identified by GC-MS. We found that the predominant constituents in Pe-W and Pe-C were methyl linoleate (60.95% vs 58.25%) and hexadecanoic acid methyl ester (10.44% vs 14.32%). Linoleic acid and its methyl esters have many beneficial effects in the prevention of atherosclerosis, cancer, hypertension and improvement of immune function (*Whelan, 2008*; *Bhattacharya et al., 2006*). It has been reported that linoleic acid or conjugated linoleic acid can inhibit the proliferation of hybridoma cells or human breast cancer MCF74 cells through induction of apoptosis or activation of p53, respectively (*Albright et al., 2005*; *Kisztelinski et al., 2006*). These studies suggest that methyl linoleate may be associated with the antitumor effects of Pe-W and Pe-C. However, other two major constituents in Pe-W and Pe-C are much different, 11-octadecenoic acid methyl ester (6.95% vs 0.73%) and methyl oleate (0% vs 15.23%). Moreover, seven components including hexadecanoic acid, hexadecanoic acid, 2-hydroxy-, methyl ester, 12-methyl-E,E-2,13-octadecadien-1-ol, Z,E-3,13-Octadecadien-1-ol, (22Z)-Cholesta-5,7,22 -trien-3-ol, desmosterol, ergosterol were found in Pe-W but not in Pe-C. Some of them may have antitumor activity, such as ergosterol, which can reduce the breast cancer cell viability through induction of apoptosis and up-regulation of Foxo3 expression (*Li et al., 2015b*). The difference of components might cause the different antitumor activities of Pe-W and Pe-C. Therefore, it is worth further isolating and identifying the constituents that cause the different antitumor activities between wild and cultivated *P. ferulae*.

Our previous study showed that PFEE-C could inhibit the growth of tumor cells through induction of apoptosis (*Wang et al., 2014*). Similarly, PFEE-W/C and their subfractions significantly induced apoptosis of HeLa cells. Furthermore, ROS production induced by Ea-W and Ea-C partially caused apoptosis. However, ROS production showed dynamically changes upon Ea-W and Ea-C treatment, which might be correlated with their antioxidant activities. The possible reason is that some components in Ea-W and Ea-C quickly induced ROS production during the first 2 h and gradually scavenged by other components in Ea-W and Ea-C from 8 to 24 h. Finally, ROS production reerupted at 48 h due to the progress of apoptosis in cells.

## CONCLUSION

These results indicated that the different extracts of wild and cultivated *P. ferulae* possessed antioxidant and antitumor activities. Although there are some differences in antioxidant and antitumor activities, cultivated *P. ferulae* could replace the wild one to a great extent. *P. ferulae* could be used to develop functional food with antioxidant and antitumor activities.

### Funding

This work was supported by the "13th Five-Year" Plan for Key Discipline Biology Bidding Project (17SDKD0202), Xinjiang Normal University to Jinyu Li and the Chinese National Natural Science Foundation Grant (31460241) and the High Level Talent Introduction Project of Xinjiang Uygur Autonomous Region to Jinyao Li. The funders had no role in study design, data collection and analysis, decision to publish, or preparation of the manuscript.

### Grant Disclosures

The following grant information was disclosed by the authors:
"13th Five-Year" Plan for Key Discipline Biology Bidding Project: 17SDKD0202.
Xinjiang Normal University to Jinyu Li and the Chinese National Natural Science Foundation: 31460241.
High Level Talent Introduction Project of Xinjiang Uygur Autonomous Region to Jinyao Li.

### Competing Interests

The authors declare that they have no competing interests.

### Author Contributions

- Yi Yang performed the experiments, analyzed the data, prepared figures and/or tables.
- Changshuang Fu performed the experiments.
- Fangfang Zhou performed the experiments.
- Xiaoyu Luo performed the experiments.
- Jinyu Li conceived and designed the experiments, analyzed the data, contributed reagents/materials/analysis tools, prepared figures and/or tables.
- Jun Zhao conceived and designed the experiments.
- Jiang He conceived and designed the experiments.
- Xiaoqin Li analyzed the data.
- Jinyao Li conceived and designed the experiments, analyzed the data, contributed reagents/materials/analysis tools, prepared figures and/or tables, authored or reviewed drafts of the paper, approved the final draft.

### Data Availability

Raw data is available in the Supplemental Files.

### Supplemental Information

Supplemental information for this article can be found online at http://dx.doi.org/10.7717/peerj.6097#supplemental-information.

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
