# Peer review of "Chemical composition, antioxidant and antitumor activities of sub-fractions of wild and cultivated Pleurotus ferulae ethanol extracts"

_PeerJ, doi:10.7717/peerj.6097_

## Round 0.1 · original submission · Major Revisions

I agree with reviewers comments that should all be carefully addressed.

Reviewer 1 ·

Basic reporting

The manuscript should be deeply revised both for English and for the description of the results appearing quite confuse; moreover many typing mistakes are present all in the text, especially in the list of references.
It is necessary to compare results with previous research, in the manuscript is not clear presented the previous reports of antioxidant capacity, such as reports by autors Alam et al : Consequence of the antioxidant activities and tyrosinase inhibitory effects of various extracts from the fruiting bodies of Pleurotus ferulae.

Experimental design

Figures are excellent designed, but according my opinion it will be better if the Table 3 and 4 will be merged. It will be easier to compare results of GC-MS analyses. It will be necessary to upload one of chromatograms.
Since that some methods are well known (Determination of polysaccharide content, Determination of polyphenol content, Determination of total flavonoid content, Reducing Power), maybe it can be more effective if you add only reference without detailed explanation and add modification.
The PCA analyses of the content of polysaccharides, flavoinoids and polyphenols, as well as results of antioxidant capacity and anticancer effect an be useful tool to explain and for connecting these results, which are really insufficient explained.

Validity of the findings

The subject of manuscript is very current and has not been sufficiently explored according to literature data.Furthermore, authors did not compared results with previous reported. These results are very notable, but the effect of measured chemical compounds on bioactive effect is not explained. Also,it is necessary deeper explanation of GC-MS results, in order to detect the difference, which can be cause for different bioactive activity of analyzed samples.

Additional comments

Introduction must to be more informative and included previous reports about chemical composition and bioactive effects of Pleurotus ferula.
Page 1, line 39-44; please delete the sentences since they are little informative in this context.
Page 2, line 57; please check abbreviation EA-C (also in further text)
Page 2, line 59-63; please replace these sentences to conclusion or delete.

Material and methods

Page 2, line 66; please add the source of cultivated and wild fungus
Page 2, line 74; it is not clear the production process of sub-fractions. What is dissolved in DMSO?
Page 5, line 152: Can you explain the sample preparation for GC-MS or add reference.
Page 5, line 164: Which program did you use for statistical analyses?

Results

The explanation of results is not in agreement with the stat analyses. The authors have serious mistakes in the part The polysaccharide, polyphenol and flavonoid contents of PFEE-W and PFEE-C and sub-fractions.
Page 7, line 170-172; Polysaccharide content in Pe-W is lower than that in Pe-C and Ea-W and Ba-W are similar with Ea-C and Ba-C, respectively.
This sentence is not correct, since that the polysaccharide content of these samples are Pe-W (9.764), Ea-W (20.36), Ba-W (40.64), PFEE-C (54.87), Pe-C (20.84), Ea-C (16.14), and Ba-C (38.51). Please check also the letter that showed significant difference. Why is significant difference marked with b for sample PFEE-W (38.46±1.005b)and not with a ?
Page 7, line 173-174. The explanation of results did not in agreement with results of the stat analyses, since that according to presented results Ea-C and BA-W are marked with some letter, thus these are not significant different.

Page 8,line 204; Why did you choose these samples (Pe-W and Pe-C) for GC-MS analyses?

Reviewer 2 ·

Basic reporting

see below

Experimental design

see below

Validity of the findings

see below

Additional comments

The authors compared extracts from WT and cultured PF and showed that there were no significant difference in inhibition of tumor cell growth, induction of apoptosis and ROS.
It is important and useful for development of nature anticancer drugs and healthy food usage. However, the Ms lacks control samples and some results and explanation is not very convincing.
Non cancer cell line should be included in all experiments.
line 149, WAS DCFH-DA directly added to the culture, or cells were harvested before they were stained? give details of the experiment
line 176-177 delete ” The results indicated that PFEE-W/C and sub-fractions might have
177 different antioxidant and antitumor activities due to their different contents of some components” due to at the moment data cant indicate anything.
The authors should do statistical analysis for data presented in Table 2 and table3

Fig 3 B, pls show summarized data in addition to representative imagesThere are many grammar errors.
The explanation for ROs change does not make sense.
Line 261-262, the authors claimed that the effect of Ea-W and Ea-C on ROS is due to their antioxidant activity. It is contradict. ROS induces apoptosis. Not because of apoptosis results in ROS increase. The authors should clarify and demonstrate it.
There are many grammar errors.

Reviewer 3 ·

Basic reporting

Minor suggestions:

1. In the Materials and Methods section under Preparation of wild and cultivated P. ferulae extractions, it states on line 67, “100 g powders of wild and cultivated P. ferulae were extracted three times using 1 L of 95% (v/v) ethanol”- done to obtain PFEE-W and PFEE-C extracts and then on line 72, “Then PFEE-W and PFEE-C were successively extracted with equal volume of petroleum ether”- where they obtain the sub-fractions. The meaning of the word “extracted” appears to be different in the 2 contexts and is confusing. The authors need to fix the wording in these lines.

2. Describe what B16, Eca-109, BCG823, and HeLa cells are in the Cell lines and cell culture part of the Materials and Methods section. For e.g. B16 mouse melanoma cells.

3. In Figure 3B, the nuclear morphology is hard to discern. However, it is better in the raw file. The authors are encouraged to make an inset within Figure 3B containing a magnified portion of the figure to facilitate easy visualization of the nuclear morphology.

Experimental design

In this manuscript titled “Chemical composition, antioxidant and antitumor activities of sub-fractions of wild and cultivated Pleurotus ferulae ethanol extracts”, Yang. Y et al. compare the bioactivities of organic extracts of wild and cultivated P. ferulae edible mushroom. This manuscript builds on earlier evidence that P. ferulae extracts possess antioxidant and antitumor properties (Alam N et al 2012, Wang W et al 2014, Choi DB et al 2004). In particular, they analyze the polysaccharide, polyphenol, and flavonoid composition in P. ferulae ethanol extracts and in the petroleum ether, ethyl acetate, and n-butanol sub-fractions of the ethanol extracts. Furthermore, the authors measure the antioxidant potential of the extracts using DPPH scavenging and reducing power assays. Ethanol extracts and sub-fractions of wild and cultivated P. ferulae display a dose-dependent increase in antioxidant activity. The extracts are then tested for antitumor activity on mouse and human tumor cell lines. Higher concentrations of wild P. ferulae extracts show more dramatic effect on cell viability than cultivated P. ferulae extracts, particularly in Eca-109 and HeLa cells. The authors provide evidence that P. ferulae extracts potentiate the release of reactive oxygen species (ROS) in the tumor cells that triggers cell death.

Some noteworthy aspects of the study:
1. Use of Cisplatin and DMSO as controls in cell viability experiments.
2. Detailed descriptions of experimental methods.
3. Analysis of apoptotic and necrotic cell populations in treated cells.
4. GC-MS analysis to determine chemical composition of petroleum ether sub-fractions of wild and cultivated P. ferulae ethanol extracts.

Validity of the findings

However, the authors need to address the following concerns to publish the manuscript:

1. According to the Preparation of wild and cultivated P. ferulae extractions in the Materials and Methods section, ethanol extracts were evaporated to remove ethanol and then further treated with petroleum ether, ethyl acetate, and n-butanol. It is not clear whether these solvents were also evaporated prior to reconstitution in DMSO. In addition, line 74 should mention that the extracts were dissolved in 100 percent DMSO.
a. If the solvents petroleum ether, ethyl acetate, and n-butanol were not evaporated, then it must be shown either by experimental evidence or literature citation whether these solvents have any effect on DPPH scavenging, reducing power, and cell viability.

2. In the Results section, Polysaccharide, polyphenol and flavonoid contents of PFEE-W and -C and subfractions, the authors state on line 171 “Polysaccharide content in Pe-W is lower than that in Pe-C and Ea-W and Ba-W are similar with Ea-C and Ba-C, respectively.”

The polysaccharide content in Pe-W is lowest compared to all other sub-fractions. The statement is not clear and does not correlate with the numbers on Table 1.

3. In Figure 1, DPPH scavenging and reducing power assays, there is no reference positive control (a known compound with antioxidant properties). It is hard to assess the importance of these results in the absence of a reference. The authors must either provide experimental data using a reference positive control or draw comparison to other compounds that have antioxidant properties in the Discussion section.

4. The authors analyze the differences in fatty acid content of petroleum ether sub-fractions of wild and cultivated P. ferulae using GC-MS and claim that the differences in the constituents may cause different antitumor activities. The authors need to explain this further in the Discussion citing studies that show antitumor activity of compounds containing various fatty acids.

5. In Figure 4A, the authors show data that Ea-W and Ea-C sub-fractions can induce ROS production in HeLa cells. The control used in these experiments is 0.4% DMSO. DMSO itself appears to induce ROS production at 8h, 12h, and 24 timepoints. The authors must add untreated cells as an additional control to show the baseline level of ROS at these timepoints. The authors are encouraged to find other literature where dynamic changes in ROS has been reported and include in the Discussion section.

---

## Round 0.2 · Minor Revisions

There are still minor issues to address. Please also, avoid to define NCTC1469 as "normal" cells, they are mouse liver cell lines (or non cancer cell lines as suggested by the reviewer).

Reviewer 3 ·

Basic reporting

Reporting has improved with the addition of correct descriptions of the experimental methods and reagents as well as detailed discussion of data.

Experimental design

Minor issues:

The authors have carried out additional experiments to include ascorbic acid as a positive control for data presented in Figure 1. However, the authors have not discussed how the Pleurotus ferulae extracts compare to the positive control. In addition, Figure 1 legend needs to include a key for "Vc".

Validity of the findings

The authors' inclusion of positive and negative controls have strengthened the validity of the results.

Additional comments

The authors have adequately addressed reviewers' concerns.

The manuscript is suitable for publication upon correction of minor issues.

---

## Round 0.3 · accepted · Accept

All the issues have been correctly addressed.

#